# Uncertainty-Aware Alignment Network for Cross-Domain Video-Text Retrieval

**Xiaoshuai Hao**[1], **Wanqian Zhang**[2]*
[1]Samsung Research China–Beijing (SRC-B)
[2]Institute of Information Engineering, Chinese Academy of Sciences
xshuai.hao@samsung.com, zhangwanqian@iie.ac.cn

## Abstract

Video-text retrieval is an important but challenging research task in the multimedia community. In this paper, we address the challenge task of Unsupervised Domain Adaptation Video-text Retrieval (UDAVR), assuming that training (source) data and testing (target) data are from different domains. Previous approaches are mostly derived from classification based domain adaptation methods, which are neither multi-modal nor suitable for retrieval task. In addition, as to the pairwise misalignment issue in target domain, i.e., no pairwise annotations between target videos and texts, the existing method assumes that a video corresponds to a text. Yet we empirically find that in the real scene, one text usually corresponds to multiple videos and vice versa. To tackle this one-to-many issue, we propose a novel method named Uncertainty-aware Alignment Network (UAN). Specifically, we first introduce the multimodal mutual information module to balance the minimization of domain shift in a smooth manner. To tackle the multimodal uncertainties pairwise misalignment in target domain, we propose the Uncertainty-aware Alignment Mechanism (UAM) to fully exploit the semantic information of both modalities in target domain. Extensive experiments in the context of domain-adaptive video-text retrieval demonstrate that our proposed method consistently outperforms multiple baselines, showing a superior generalization ability for target data.

## 1 Introduction

With the exponential growth of user-generated videos on the Internet, cross-modal retrieval between video data and natural language descriptions, known as video-text retrieval, has attracted much attention. The goal of this task is to retrieve videos which are semantically related to a given text query and vice versa. Generally, the de facto paradigm of video-text retrieval task is to learn high-level video and text embeddings respectively by off-the-shelf feature extractors. To further measure cross-modal semantic similarities between videos and texts, corresponding embeddings should be aligned and projected into a joint embedding space [3, 10, 27, 41, 44, 55, 56, 61], where the semantically-similar texts and videos are much closer to each other and vice versa.

With the advent of deep learning, video-text retrieval has shown promising performance in recent years. Despite the great success, most existing works of this task are supervised, which typically train models on a large number of aligned video-text pairs, making an assumption that training and testing data are drawn from the same distribution. If this assumption does not hold, traditional cross-modal retrieval methods may suffer a performance drop at the evaluation. To alleviate this, recently proposed methods of Unsupervised Domain Adaptation Video-text Retrieval (UDAVR) have attracted much attention, wherein no *identical label set* for source and target domains exists. By *identical* we denote that the label set of source data is exactly the same as that of target data. For instance, given a

---

*Corresponding author.

37th Conference on Neural Information Processing Systems (NeurIPS 2023).

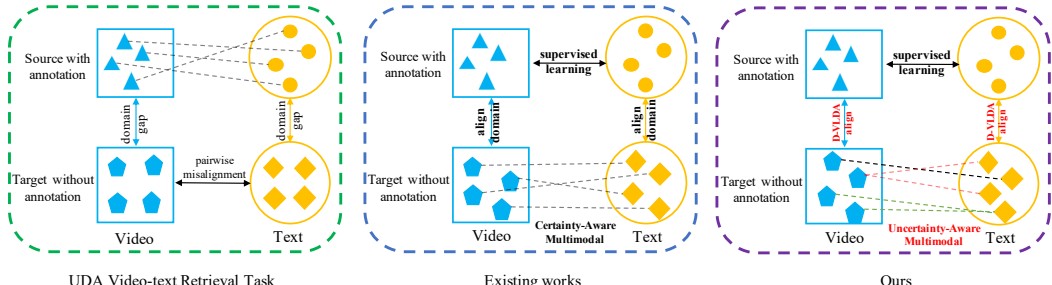

Figure 1: Illustration of the UDA Video-text Retrieval task, existing works and the proposed method. The proposed method utilizes Distribution-based Vision-Language Domain Adaptation(D-VLDA) in domain gap and uncertainty-aware multimodal alignment mechanism in the target domain.

video-text pair labeled as *play basketball* in source domain, *identical label set* requires the existence of target pair which also labeled as it. This can barely be guaranteed due to the diversities between different datasets, such as scenarios, video scenes and lengths. Thus, the only supervision of UDAVR is the semantic relationship in source dataset, i.e., whether one source video and one source text are matched, which is also the general setting in other UDA cross-modal tasks [4, 8, 60, 62].

To alleviate the domain shift problem, recently proposed methods of Unsupervised Domain Adaptation (UDA) has attracted much attention, such as image classification [32, 52], autonomous driving [50, 51] and video-based action recognition [45, 48]. However, these methods are originally designed for classification tasks, which are not suitable for the video-text retrieval task. To that end, several pioneering UDAVR methods have been recently proposed [6, 13, 22, 37]. They try to tackle the lack of target domain supervision by directly minimizing the distribution discrepancy [13], distilling knowledge from the source domain [6], introducing pre-defined prototype assignments [37], or adaptively aligning the video-text pairs which are more likely to be relevant in target domain [22]. Despite the success on improving the retrieval performance of target data, they overlook the challenging one-to-many relationship issue in target domain, i.e., existing methods assume that one text in the target domain corresponds to only one video and vice versa. However, in real scenes, one query text usually corresponds to multiple videos or one query video corresponds to multiple texts.

As illustrated in Figure 1, previous methods are derived from classification based domain adaptation methods, which are neither multi-modal nor suitable for retrieval task. In addition, for the pairwise misalignment issue in target domain, the existing methods assume that a video corresponds to a text. This will inevitably mix target videos and texts together. Consequently, indiscriminative target video and text features will be introduced, leading to less generalized model and poor performance on target data. This is rather challenging and thus becomes the motivation of our work. In this paper, we propose a novel method named Uncertainty-aware Alignment Network (UAN). We first introduce the cross-modal semantic embedding to generate discriminative source features in a joint embedding space, where semantically relevant pairs should lie close together and vice versa. To alleviate the domain shift, we further utilize the Distribution-based Vision-Language Domain Adaptation(D-VLDA) to minimize of distribution shifts between source and target domains. To tackle the *one-to-many* in target domain, we propose a simple yet effective Uncertainty-aware Alignment Mechanism (UAM), which fully exploits the semantic information of both modalities in target domain. Extensive experiments on several benchmarks demonstrate the superiority of our method.

The contributions of this paper are mainly threefold:

- For the challenging Unsupervised Domain Adaptation Video-text Retrieval (UDAVR) task, we propose a simple yet effective Uncertainty-aware Alignment Network (UAN), which fully exploits the semantic information of both modalities in target domain.

- To tackle the *one-to-many* in target domain, the proposed Uncertainty-aware Alignment Mechanism (UAM) tries to utilize the multi-granularity relationships between each target video and text to ensure the discriminability of target features.

- Compared with the state-of-the-art methods, UAN achieves 15.47% and 11.57% relative improvements on R@1 under the setting of TGIF→MSRVTT and MSRVTT→TGIF respectively, demonstrating the superiority of our method.

## 2 Related Work

**Video-Text Retrieval.** There are two main types of video-text retrieval approaches: single-modal based and multi-modal based. Initially, single-modal based approaches extend image-text retrieval methods. HGR [7] suggests a Hierarchical Graph Reasoning model that separates text into a hierarchical semantic graph with different levels. Additionally, GPO [5] offers an innovative Generalized Pooling Operator that adapts itself to the best pooling strategy for various baselines. More recently, the widely adopted Contrastive Language-Image Pretraining(CLIP) [2]model is now commonly used in video-text retrieval [1, 18, 20, 30, 34, 36, 43]. CLIP4Clip [42] transfers the knowledge of large-scale image-text pre-training to the video-text retrieval task through fine-tuning and explores three similarity calculation mechanisms based on the pre-trained CLIP model. Similarly, CLIP2video [14] concentrates on the spatial semantics derived from the CLIP model. Another research direction has emphasized capturing multi-modal representations using various experts, which are then combined to derive the final video features. For instance, CE [35] uses video features from all modalities to encode a video, while T2VLAD [54] concentrates on automatically learning text-and-video semantic topics and emphasizing the significance of local semantic alignment between them. On the other hand, MMT [15] utilizes multi-modal information extracted by seven pretrained experts but only fuses them in a straightforward manner without specific guidance. *In contrast to these methods, we explore the video-text retrieval task through the lens of unsupervised domain adaptation.*

**Unsupervised Domain Adaptation for Video-Text Retrieval.** UDA transfers knowledge from a labeled source domain to an unlabeled target domain [12, 32, 38, 40, 49], which is realized by training the model with both labeled source data and unlabeled target data. Recently, some cross-modal tasks resort to UDA and try to utilize the unpaired data in target domain, such as image captioning [8, 60, 62] and VQA [4]. DCKT [26] focuses on UDA image-text retrieval and transfers knowledge from a large dataset to promote the model performance on small dataset. *However, these UDA methods are originally designed for classification tasks, which might not be suitable for the video-text retrieval.* To our knowledge, there are only a few explorations of the Unsupervised Domain Adaptation for Video-Text Retrieval task [6, 13, 22, 37]. MAN [13] proposes a Multi-level Alignment Network to alleviate different gaps in UDAVR task. CAPQ [6] introduces the inaugural benchmark for unsupervised domain adaptation in video-text retrieval. ACP [37] proposes a new adaptive cross-modal prototypes framework to minimizing uni-modal and cross-modal shifting between the source and target domains. DADA [22] introduces the Dual Alignment Consistency (DAC) mechanism adaptively aligns the most similar videos and texts in target domain. Our approach diverges from previous works in two principal ways. (1) Previous approaches are mostly derived from classification based domain adaptation methods to solve domain discrepancy problem, which are neither multi-modal nor suitable for retrieval task. (2) Prior works have not fully leveraged the semantic connections between videos and text in target domains, assuming that a video corresponds to a text. However, in the real scene, one text usually corresponds to multiple videos and vice versa ( as illustrated in Figure 3). This problem is the primary motivation for our paper.

## 3 Methodology

### 3.1 Preliminaries

**Problem Definition.** For notational clarity, we first introduce the symbols and definitions used throughout this paper. Formally, assume that we have a set of samples in source domain $\left\{ (\mathcal{V}^s, \mathcal{T}^s) = (v_i^s, t_i^s)_{i=1}^{n_s} \right\}$, where $n_s$ indicates the number of video-text pairs. Similarly, there also exists a set of samples in target domain $\left\{ \mathcal{V}^t = \{v_i^t\}_{i=1}^{n_t}, \mathcal{T}^t = \{t_j^t\}_{j=1}^{n_t} \right\}$ with two collections of $n_t$ videos $\mathcal{V}^t$ and texts $\mathcal{T}^t$, respectively. Note that the target videos and texts are *unpaired*, which means the supervised information is not available in target domain. In other words, we have no idea *whether one target video-text pair is semantically relevant or not* during the training procedure. The objective of Unsupervised Domain Adaptation Video-Text Retrieval (UDAVR) is to enhance the model's ability to generalize to the target domain by leveraging information from the source domain. The overall framework of our method is illustrated in Figure 2.

**Semantic Embedding Learning.** Using the latest baseline in video-text retrieval, as described in [5], we employ a video encoder $\varphi(\cdot)$ and a text encoder $\psi(\cdot)$ to generate a joint embedding space for a given video-text pair, consisting of a video sample $v$ and a text description $t$. For both source and

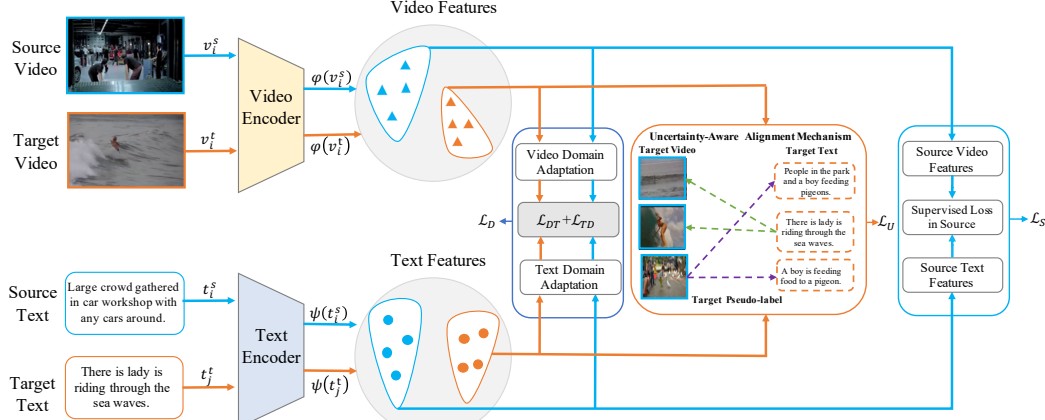

Figure 2: The overall framework of UAN. Semantic Embedding Learning generates discriminative source features in a joint embedding space ($\mathcal{L}_S$). Distribution-based Vision-Language Domain Adaptation (D-VLDA) is proposed to alleviate the domain discrepancy problem in both modalities($\mathcal{L}_D$). Uncertainty-Aware Alignment Mechanism (UAM) is proposed to dig in uncertainty-aware multimodal relationships in the target domain($\mathcal{L}_U$).

target data, we share the same feature extracting modules to map them into the common space. Such a design can be interpreted as to some extent *transfer the knowledge learned in the source domain to the target domain*. If the text effectively describes the video, the visual embedding $\varphi(v) \in \mathbb{R}^M$ and the text embedding $\psi(t) \in \mathbb{R}^M$ exhibit semantic relevance, with $M$ representing the dimension in the shared space. In the source domain, we use the symmetric cross-entropy loss to guide the semantic alignment learning. Given a batch of B text-video pairs, the model updates its parameters through maximizing the sum of the main diagonal of a B × B similarity matrix. Normally, we represent the cross-entropy loss in text-to-video directions in Eq. 1 and represent the cross-entropy loss in video-to-text directions in Eq. 2. The similarity of the text-video is calculated as the inner production of $t_i$ and $v_i$ in Eq.3. The above formula can be formulated as:

$$\mathcal{L}_{t2v} = -\frac{1}{B} \sum_i^B log \frac{exp\left(s\left(t_i, v_i\right)\right)}{\sum_j^B exp\left(s\left(t_i, v_j\right)\right)} , \qquad (1)$$

$$\mathcal{L}_{v2t} = -\frac{1}{B} \sum_i^B log \frac{exp\left(s\left(v_i, t_i\right)\right)}{\sum_j^B exp\left(s\left(v_i, t_j\right)\right)} , \qquad (2)$$

$$s\left(t_i, v_i\right) = \langle t_i, v_i \rangle = \langle \psi\left(t_i\right) \cdot \varphi\left(v_i\right) \rangle , \qquad (3)$$

where $\psi\left(t_i\right)$ and $\varphi\left(v_i\right)$ are the corresponding mapped features. Formally, the symmetric cross-entropy loss $\mathcal{L}_S$ is the sum of text-to-video loss $\mathcal{L}_{t2v}$ and video-to-text loss $\mathcal{L}_{v2t}$, formulated as:

$$\mathcal{L}_S = \mathcal{L}_{t2v} + \mathcal{L}_{v2t} . \qquad (4)$$

### 3.2 Distribution-based Vision-Language Domain Adaptation

In light of the video-text symmetric cross-entropy loss, the representations of unimodal encoders can be effectively aligned, which enables the generation of more informative samples. Thus, we can obtain discriminative and modality-invariant features of source videos and texts in the joint embedding space. However, this discriminability can not generalize well to the target domain, due to the inevitable *domain discrepancy* problem.

To alleviate this, existing works typically utilize classification domain adaptation (DA) methods to shift the feature distributions in different domains as close as possible for each modality, such as IDM [11] and MMD [38]. *However, these methods are originally designed for classification tasks, which is not suitable for the video-text retrieval task.* Thus, we propose Distribution-based Vision-Language Domain Adaptation, called **D-VLDA**, to relieve the divergence of domain statistics, thus the distribution shifts can be significantly diminished, improving the generalization of learned model on out-of-distribution target domain. The basic idea is that the moment calculated on different

domains are the same if the distributions of source and target domains are identical. Specifically, for two Gaussian distributions $\mathcal{N}(\mu_1, \Sigma_1)$ and $\mathcal{N}(\mu_2, \Sigma_2)$, their 2-Wasserstein distance is defined as:

$$D_{2W} = \|\mu_1 - \mu_2\|_2^2 + \text{Tr}\left(\Sigma_1 + \Sigma_2 - 2\left(\Sigma_1^{1/2}\Sigma_2\Sigma_1^{1/2}\right)^{1/2}\right), \tag{5}$$

In our modeled distributions, $\Sigma_1$ and $\Sigma_2$ are both diagonal matrices, indicating $\Sigma_1^{1/2}\Sigma_2\Sigma_1^{1/2} = \Sigma_1\Sigma_2$. The above formula can be rewritten as:

$$D_{2W} = \|\mu_1 - \mu_2\|_2^2 + \text{Tr}\left(\left(\Sigma_1^{1/2} - \Sigma_2^{1/2}\right)^2\right),$$
$$= \|\mu_1 - \mu_2\|_2^2 + \|\sigma_1 - \sigma_2\|_2^2, \tag{6}$$

where $\sigma$ refers to a standard deviation vector. In our task, the feature distribution consists of two folds: video domain adaptation module and text domain adaptation module. Therefore, we propose two domain alignment loss functions $L_{DT}$ and $L_{DV}$ as follows:

$$L_{DT} = a \cdot D_{2W}\left(\mathcal{V}^s, \mathcal{V}^t\right) + b, \tag{7}$$

$$L_{DV} = a \cdot D_{2W}\left(\mathcal{T}^s, \mathcal{T}^t\right) + b, \tag{8}$$

$$L_D = L_{DT} + L_{DV}. \tag{9}$$

where $a$ is a negative scale factor since similarity is inversely proportional to the distance, and $b$ is a shift value. Thus, by reducing the distance between two distributions based on Eq. 9, we align the feature distributions of the same modal in different domains.

### 3.3 Uncertainty-aware Alignment Mechanism

Besides alleviating the domain discrepancy, we also need to bridge the semantic gap between videos and texts in the target domain, which is the key to the standard UDAVR task. Specifically, given the target set $\left\{\mathcal{V}^t = \{v_i^t\}_{i=1}^{n_t}, \mathcal{T}^t = \{t_j^t\}_{j=1}^{n_t}\right\}$, pairwise information is not available in the target domain. This means that the target videos and texts are *unpaired*, and as a result, the issue of *one-to-many* arises in the target domain. However, existing method [22] assumes that a video corresponds to a text, whereas in the real scene, one text usually corresponds to multiple videos and vice versa (as illustrated in Figure 3). To alleviate this, we propose a simple yet effective Uncertainty-Aware Alignment Mechanism (UAM). The UAM tries to utilize the multi-granularity relationships between each target video and text to guide the unsupervised learning in target dataset. We calculate the similarities $(S_{v_i^t}^{\mathcal{T} \to \mathcal{V}})$ between a target text $t_i^t$ and all target videos, and the similarities $(S_{t_i^t}^{\mathcal{V} \to \mathcal{T}})$ between a target video $v_i^t$ and all target texts, which can be defined as follows:

$$\mathbf{S}_{v_i^t}^{\mathcal{T} \to \mathcal{V}} = [S_{t_i^t v_1^t}, S_{t_i^t v_2^t}, ..., S_{t_i^t v_j^t}, ..., S_{t_i^t v_{n_t}^t}]. \tag{10}$$

$$\mathbf{S}_{t_i^t}^{\mathcal{V} \to \mathcal{T}} = [S_{v_i^t t_1^t}, S_{v_i^t t_2^t}, ..., S_{v_i^t t_j^t}, ..., S_{v_i^t t_{n_t}^t}]. \tag{11}$$

$$\mathcal{I}\left(v_{i*}^t, t_{j*}^t\right) = \begin{cases} 1, & \text{if } v_{i*}^t \text{ and } t_{j*}^t \text{ are reciprocal } \textit{TOP-K} \text{ similar}, \\ 0, & \text{otherwise}. \end{cases} \tag{12}$$

Note that this requires $v_{i*}^t$ and $t_{j*}^t$ to be the reciprocal *TOP-K* similar of each other, indicating a truly aligned (or positive) pair. To that end, we can obtain $n_p$ positive pairs in one batch from target dataset, denoted as $\left\{(\mathcal{V}^p, \mathcal{T}^p) = (v_i^p, t_i^p)_{i=1}^{n_p}\right\}$, where $(v_i^p, t_i^p) = (v_{i*}^t, t_{j*}^t)$. To improve the accuracy of aligned pairs, we introduce $T$ as a threshold for sorting the similarities of all pairs in a batch in descending order and selecting the $T$-th value, which corresponds to the top $T$ most similar pairs. Our rationale is that a truly positive video-text pair should not only be the most similar to each other but also have a relatively high similarity score compared to all the misaligned pairs. Formally, the uncertainty-aware alignment matching operation of Eq. 12 can be revised as:

$$\mathcal{I}\left(v_{i*}^t, t_{j*}^t\right) = \begin{cases} 1, & \text{if } S_{v_i^p t_i^p} \geq S_T, \\ 0, & \text{otherwise}. \end{cases} \tag{13}$$

where $S_T$ denotes the top $T$-th value in one batch and $S_T \in [\mathbf{S}_{v_i^t}^{\mathcal{V} \to \mathcal{T}}; \mathbf{S}_{t_j^t}^{\mathcal{T} \to \mathcal{V}}]_{des} = [S_1, S_2, ..., S_T, ..., S_{B^2}]_{des}$. $[\mathbf{S}_{v_i^t}^{\mathcal{V} \to \mathcal{T}}; \mathbf{S}_{t_j^t}^{\mathcal{T} \to \mathcal{V}}]_{des}$ denotes re-ranking in a descending order (from similar to dissimilar). The combination of uncertainty-aware alignment matching and the similarity

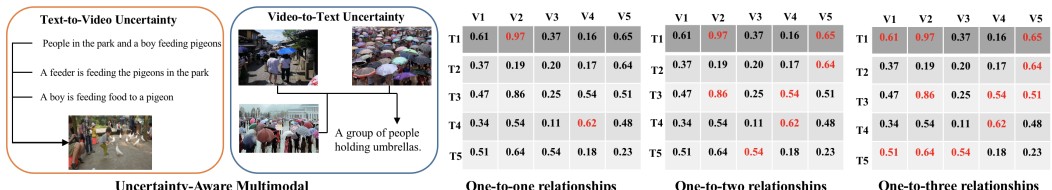

Figure 3: Illustration of Uncertainty-Aware Alignment Mechanism (UAM). If $v_{i*}^t$ and $t_{j*}^t$ to be the reciprocal *TOP-K* similar of each other, indicating a truly aligned (or positive) pair.

threshold $T$ contributes to the alignment of target videos and texts in a mutually-promoted manner. By using these self-discovered matching pairs, we can approach the pairwise misalignment issue as a fully supervised problem. We can define the uncertainty-aware alignment loss as:

$$
\begin{aligned}
\mathcal{L}_U = -\frac{1}{B}(&\sum_i^B log \frac{exp\left(s\left(t_i^p, v_i^p\right)\right)}{\sum_j^B exp\left(s\left(t_i^p, v_j^p\right)\right)} \\
&+ \sum_i^B log \frac{exp\left(s\left(v_i^p, t_i^p\right)\right)}{\sum_j^B exp\left(s\left(v_i^p, t_j^p\right)\right)}).
\end{aligned}
\tag{14}
$$

Therefore, the full training objective is given by:

$$
\mathcal{L} = \mathcal{L}_S + \lambda_1 \mathcal{L}_D + \lambda_2 \mathcal{L}_U,
\tag{15}
$$

where $\lambda_1$ and $\lambda_2$ are hyper parameters for balancing these terms.

## 4 Experiments

### 4.1 Experimental Setting

**Datasets.** We use existing datasets across three domains to explore the UDAVR task, i.e., a comprehensive evaluation benchmark by combining three popular datasets: MSR-VTT (Mt)[59], MSVD (Md)[19], and TGIF (Tf)[33]. We assume that data from different datasets are from different domains, as they often have dissimilar data distributions and representations.

**Evaluation Metrics.** To evaluate the performance of our video-text retrieval, we utilize standard retrieval metrics, as outlined in previous studies ([7, 21, 23, 24, 25]). Our analysis is based on rank-based metrics, including R@K (the higher, the better) and Median Rank, i.e., MR, (the lower, the better).

**Implementation Details.** To ensure a fair comparison, we employ the identical architecture for the video and text encoders as used in GPO [5], which is currently the leading baseline for video-text retrieval. The length of shared embedding $M$ is set to 1024. Moreover, we adopt the Adam optimizer for all our experiments. During the training procedure, we set the batch size to 64, and utilize a step-decayed learning rate with initialization value 0.0001. All experiments are conducted five times for the average performance on a 2080Ti GPU server. For all our experiments, we set $a$ to -0.005 and $b$ to 6. The hyper parameters $\lambda_1$, $\lambda_2$, $K$ and $T$ of the overall objective function loss is discussed extensively in section 4.3. We set the max epochs as 100, and early stop occurs if the validation performance does not improve in ten consecutive epochs.

### 4.2 Performance Comparison and Analysis

We compare our method with several state-of-the-art baselines across three categories, i.e., Source Only, DA methods and UDAVR methods. As a lower bound, we include the non-adapted Source Only results, which directly applies the model trained on the source domain to the target domain. We also implement five classification-based (i.e. typical) DA methods and modify them for the UDAVR task, including MMD [38], CORAL [49], DANN [16], IDM [11] and SCDA [32]. Moreover, we compare to four recently proposed UDAVR methods, i.e., MAN [13], CAPQ [6], ACP [37] and DADA [22]. For fairness, all methods adopt the same features and the backbone network as [5].

Table 1: Comparison with different baselines.

| Method | | Tf→Mt | | | Mt→Tf | | | Tf→Md | | | Md→Tf | | | Mt→Md | | | Md→Mt | | |
|---|---|---|---|---|---|---|---|---|---|---|---|---|---|---|---|---|---|---|---|
| | | R1↑ | R10↑ | MR↓ | R1↑ | R10↑ | MR↓ | R1↑ | R10↑ | MR↓ | R1↑ | R10↑ | MR↓ | R1↑ | R10↑ | MR↓ | R1↑ | R10↑ | MR↓ |
| Source Only | | 2.69 | 13.63 | 144 | 6.30 | 25.43 | 60 | 9.39 | 37.77 | 20 | 3.80 | 16.99 | 102 | 15.02 | 46.96 | 12 | 2.50 | 13.27 | 136 |
| (a) | MMD [38] | 2.68 | 13.59 | 135 | 6.77 | 27.11 | 54 | 9.11 | 36.11 | 23 | 3.50 | 16.28 | 119 | 15.31 | 47.65 | 12 | 2.62 | 13.18 | 136 |
| | CORAL [49] | 2.74 | 14.07 | 128 | 6.56 | 26.49 | 52 | 9.44 | 37.87 | 21 | 3.65 | 17.34 | 108 | 15.65 | 49.43 | 11 | 2.65 | 13.34 | 138 |
| | DANN [16] | 2.76 | 13.94 | 127 | 6.86 | 27.17 | 48 | 9.27 | 38.00 | 20 | 3.74 | 16.72 | 103 | 15.67 | 48.67 | 11 | 2.62 | 13.17 | 134 |
| | IDM [11] | 2.59 | 13.11 | 149 | 7.12 | 25.35 | 60 | 8.05 | 35.51 | 23 | 3.24 | 15.78 | 120 | 13.96 | 47.77 | 12 | 2.54 | 12.39 | 165 |
| | SCDA [32] | 2.79 | 14.22 | 130 | 6.92 | 26.70 | 53 | 9.84 | 37.11 | 22 | 3.30 | 17.02 | 108 | 15.64 | 48.65 | 11 | 2.55 | 12.98 | 138 |
| (b) | MAN [13] | 2.53 | 12.98 | 144 | 6.42 | 25.96 | 63 | 8.84 | 37.06 | 21 | 3.06 | 16.31 | 119 | 15.05 | 48.51 | 11 | 2.40 | 12.00 | 174 |
| | CAPQ [6] | 3.46 | 17.02 | 110 | 7.33 | 25.64 | 62 | 9.30 | 37.97 | 21 | 3.97 | 17.75 | 113 | 15.66 | 49.08 | 11 | 3.35 | 15.47 | 158 |
| | ACP [37] | 4.41 | 21.72 | 64 | 7.83 | 26.72 | 50 | 12.09 | 41.38 | 18 | 5.12 | 21.46 | 82 | 17.87 | 54.34 | 8 | 5.90 | 25.68 | 54 |
| | DADA [22] | 5.30 | 24.54 | 50 | 8.21 | 28.97 | 45 | 14.34 | 48.77 | 11 | 6.03 | 22.52 | 78 | 18.97 | 57.93 | 7 | 6.40 | 27.61 | 42 |
| Ours | **UAN** | **6.12** | **27.23** | **40** | **9.16** | **31.06** | **37** | **15.15** | **49.34** | **10** | **6.51** | **23.93** | **69** | **20.25** | **60.23** | **5** | **6.52** | **28.15** | **40** |

Table 2: Effect of $\mathcal{L}_D$ and $\mathcal{L}_U$.

| Method | Tf→Mt | | | Mt→Tf | | |
|---|---|---|---|---|---|---|
| | R1↑ | R10↑ | MR↓ | R1↑ | R10↑ | MR↓ |
| Source Only | 2.69 | 13.63 | 144 | 6.30 | 25.43 | 60 |
| UAN(w/o $\mathcal{L}_D$) | 3.63 | 18.12 | 79 | 6.72 | 26.76 | 50 |
| UAN(w/o $\mathcal{L}_U$) | 3.87 | 18.45 | 76 | 6.91 | 27.01 | 48 |
| **UAN(full)** | **6.12** | **27.23** | **40** | **9.16** | **31.06** | **37** |

Table 3: Analysis on different alignment mechanisms.

| Method | Tf→Mt | | | Mt→Tf | | |
|---|---|---|---|---|---|---|
| | R1↑ | R10↑ | MR↓ | R1↑ | R10↑ | MR↓ |
| UAN(w/ DAC) | 5.43 | 25.63 | 48 | 8.44 | 29.12 | 42 |
| UAN(w/ UAM) | 5.76 | 26.16 | 43 | 8.73 | 29.84 | 40 |
| **UAN(full)** | **6.12** | **27.23** | **40** | **9.16** | **31.06** | **37** |

Table 1 shows that: (1) As the lower bound, Source Only achieves the worst performance, which identifies the existed domain shift problem. Compared with Source Only, our method achieves 3.43% and 2.86% absolute performance improvements on R@1 under the setting of Tf→Mt and Mt→Tf respectively, indicating the huge domain gap and the efficacy of UAN. (2) Traditional DA methods in setting (a) are ineffective for the UDAVR task, which can only slightly outperform the Source Only baseline. The reason is that these methods are originally designed for classification tasks, which might not be suitable for the video-text retrieval. (3) Our method consistently outperforms other UDAVR methods in setting (b) on all adaptation directions across the three datasets, which demonstrates the effectiveness of distribution-based vision-language domain adaptation and uncertainty-aware alignment mechanism. Compared with the SOTA method DADA, UAN achieves 15.47% and 11.57% relative improvements on R@1 under the setting of Tf→Mt and Mt→Tf respectively.

## 4.3 Ablation Studies

**Training process of different ablation models.** To systematically evaluate the effectiveness of each module of our proposed UAN, we train the model by removing each component solely and present the results in Table 2. In the main ablation study, we design the following ablation models: (1)**UAN(w/o $\mathcal{L}_D$)** : we remove the distribution-based vision-language domain adaptation module (Eq. 9) from UAN; (2)**UAN(w/o $\mathcal{L}_U$)**:we remove the uncertainty-aware alignment mechanism module (Eq. 14) from UAN; (3)**UAN(full)**:our full UAN model. The results of UAN(w/o $\mathcal{L}_D$) and UAN(w/o $\mathcal{L}_U$) are inferior to the full UAN method, verifying the effectiveness of distribution-based vision-language domain adaptation and uncertainty-aware alignment mechanism.

**Analysis on uncertainty-aware alignment mechanisms.** We conduct a comprehensive analysis of several alignment mechanisms to explore the efficacy of uncertainty-aware alignment mechanisms(UAM), and present the findings in Table 3. Specifically, (1)**UAN(w/ DAC)**: we use dual alignment consistency mechanism adaptively aligns the most similar videos and texts in target domain. (2)**UAN(w/ UAM)**: we use UAM to dig in uncertainty-aware multimodal relationships in the target domain. (3)**UAN(full)**: our full UAN model with the threshold $T$ . The results of UAN(w/ DAC) are worse than the UAN(w/ UAM), demonstrating that the uncertainty-aware alignment mechanism is superior to dual alignment consistency mechanism. Meanwhile, the result of UAN(w/ UAM) is also inferior, which proves that the constraint on high similarities of truly aligned pairs is effective. Furthermore, Figure 4 shows the effect of threshold $K$ in UAM within one batch during the training procedure. Our method consistently performs well and achieves the best when $K = 2$.

**Analysis on different DA methods.** We conduct extensive ablation studies with several state-of-the-art domain adaptation methods to verify the effectiveness of Distribution-based Vision-Language Domain Adaptation(D-VLDA). We implement three classification-based (i.e., typical) DA methods

Table 5: Generalization to different video-text retrieval methods.

| Method | | Tf→Mt | | | Mt→Tf | | |
|---|---|---|---|---|---|---|---|
| | | R1↑ | R10↑ | MR↓ | R1↑ | R10↑ | MR↓ |
| (a) | HGR [7] | 2.20 | 11.98 | 154 | 5.87 | 22.10 | 72 |
| | **HGR + UAN** | **4.52** | **21.32** | **86** | **8.43** | **29.23** | **45** |
| | GPO [5] | 2.69 | 13.63 | 144 | 6.30 | 25.43 | 60 |
| | **GPO + UAN** | **6.12** | **27.23** | **40** | **9.16** | **31.06** | **37** |
| (b) | CE [35] | 2.93 | 14.7 | 122 | 6.50 | 26.23 | 56 |
| | **CE + UAN** | **6.23** | **27.25** | **41** | **9.32** | **32.41** | **34** |
| | MMT [15] | 4.20 | 22.30 | 78 | 7.32 | 31.46 | 30 |
| | **MMT + UAN** | **6.53** | **28.32** | **34** | **9.63** | **38.89** | **19** |
| (c) | CLIP4CLIP [42] | 7.20 | 28.50 | 35 | 10.43 | 38.16 | 26 |
| | **CLIP4CLIP + UAN** | **9.32** | **37.85** | **22** | **13.55** | **47.33** | **15** |
| | CLIP2Video [14] | 7.80 | 31.50 | 31 | 11.21 | 39.48 | 23 |
| | **CLIP2Video + UAN** | **9.72** | **38.43** | **17** | **14.23** | **47.87** | **12** |

Table 6: The results of image-text retrieval.

| Method | | Open Narr→ COCO Narr | | | COCO→ COCO Narr | | |
|---|---|---|---|---|---|---|---|
| | | R1↑ | R10↑ | MR↓ | R1↑ | R10↑ | MR↓ |
| (a) | SCAN [29] | 17.4 | 52.6 | 9 | 22.3 | 72.98 | 5 |
| | VSRN [31] | 19.6 | 54.7 | 7 | 25.1 | 75.4 | 4 |
| | CE [35] | 19.6 | 56.4 | 7 | 24.5 | 75.8 | 4 |
| (b) | CDAN [39] | 20.6 | 59.2 | 6 | 22.2 | 73.3 | 5 |
| | CORAL [49] | 19.4 | 58.3 | 7 | 25.4 | 74.6 | 4 |
| | DANN [16] | 19.0 | 58.4 | 7 | 24.8 | 76.8 | 4 |
| | MMD [38] | 17.3 | 50.8 | 9 | 22.6 | 72.0 | 5 |
| | OT [57] | 20.3 | 57.1 | 8 | 25.0 | 75.6 | 4 |
| (c) | MAN [13] | 20.4 | 57.3 | 8 | 25.6 | 75.8 | 4 |
| | CAPQ [6] | 21.8 | 57.4 | 7 | 26.5 | 76.4 | 4 |
| | ACP [37] | 22.3 | 57.9 | 6 | 27.3 | 77.9 | 4 |
| | DADA [22] | 22.9 | 58.3 | 5 | 28.1 | 78.3 | 4 |
| Ours | **UAN** | **23.6** | **59.2** | **4** | **29.5** | **79.5** | **3** |

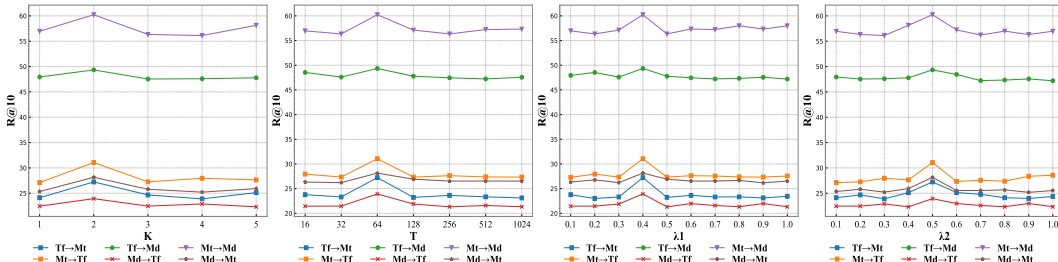

Figure 4: Analysis of hyper parameters, i.e., $K$, $T$, $\lambda_1$, and $\lambda_2$, with the R@10 retrieval performance across different domain adaptation directions.

and modify them for the UDAVR task, including MMD [38], CORAL [49] and GRL [17]. We also evaluate with TPN [46], CDD [28] and MSTN [58], which are all recently proposed DA methods based on conditional alignment. Table 4 shows how the proposed D-VLDA method performs when combined with different DA components. The proposed D-VLDA achieves significant improvements compared with classification-based and conditional alignment DA methods, which indicates that the D-VLDA module plays an essential role in the UDA video-text retrieval task.

**Generalization to different cross-modal methods.**
As shown in Table 5, we implement several state-of-the-art video-text retrieval methods and the corresponding combinations with our UAM. Clearly, our method consistently improve the performances on target domain when combined with original baselines. Moreover, we thereby generalize our UAN to the image-text retrieval scenario, to further show its effectiveness on tackling cross-modal retrieval task. For image-text retrieval, we use three datasets: COCO [9], COCO Narratives [47] and Open Narratives [47] to evaluate the generalization of the proposed method. As reported in Table 6, our

Table 4: Analysis on different DA methods.

| Method | Tf→Mt | | | Mt→Tf | | |
|---|---|---|---|---|---|---|
| | R1↑ | R10↑ | MR↓ | R1↑ | R10↑ | MR↓ |
| Source Only | 2.69 | 13.63 | 144 | 6.30 | 25.43 | 60 |
| UAN(w/ MMD [38]) | 5.62 | 25.56 | 45 | 8.36 | 29.61 | 41 |
| UAN(w/ CORAL [49]) | 5.64 | 25.58 | 44 | 8.38 | 29.64 | 41 |
| UAN(w/ GRL [17]) | 5.66 | 25.67 | 44 | 8.43 | 29.71 | 40 |
| UAN(w/ TPN [46]) | 5.72 | 25.72 | 43 | 8.46 | 29.75 | 40 |
| UAN(w/ CDD [28]) | 5.75 | 25.73 | 43 | 8.56 | 29.82 | 39 |
| UAN(w/ MSTN [58]) | 5.76 | 25.82 | 42 | 8.58 | 29.91 | 39 |
| **UAN(w/ D-VLDA)** | **6.12** | **27.23** | **40** | **9.16** | **31.06** | **37** |

method outperforms latest image-text retrieval approaches, i.e.,SCAN [29],VSRN [31] and CE [35]. Obviously, our method can be directly utilized in unsupervised domain adaptation video/image-text retrieval task, demonstrating the generalization ability of our method.

**Analysis on hyperparameters.** We conduct experiments under all the domain adaptation settings and present the sensitivity of hyper parameters in Figure 4. By altering the values within a feasible range while leaving the other hyper parameters as the corresponding default values, we can have a thorough understanding on how exactly each component contributes to the overall performance. As can be seen, within a wide range of $\lambda$ in [0.1, 1.0], the performance only varies in a small range, indicating the robustness to different choices of $\lambda$. Similarly, when progressively increasing threshold $T$ in UAM from 16 to 1,024, our method consistently performs well and achieves the best when $T = 64$ under $K = 2$ setting. Thus, we set $\lambda_1 = 0.4$, $\lambda_2 = 0.5$ and $T$ to 64 under all settings.

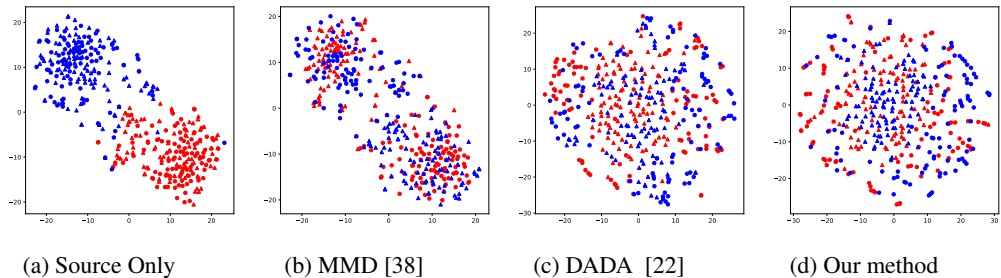

| (a) Source Only | (b) MMD [38] | (c) DADA [22] | (d) Our method |

Figure 5: The t-SNE visualizations of (a) Source Only, (b) MMD, (c) DADA and (d) Our method.

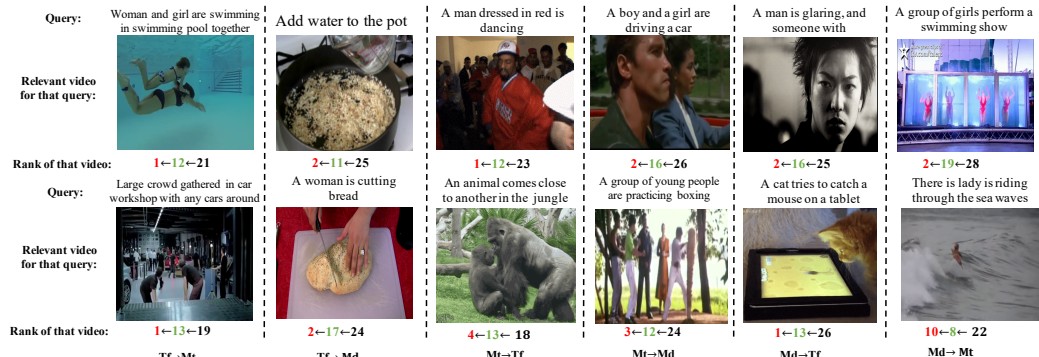

Figure 6: Visualizations of query texts and corresponding videos along with the ranking changes in A←B←C, where A denotes the rank of UAN, B the ACP method and C the Source Only.

**Feature visualizations.** We randomly choose 100 pairs in source and target domain respectively, and show the t-SNE [53] visualizations of Source Only, MMD, DADA and our method under Tf→Mt setting in Figure 5. Blue/red denotes source/target domain, while circles/triangles denote videos/texts. Clearly, Source Only method performs the worst as the blue and red features fail to mix up due to the large domain gaps, whereas (b) and (c) can both to some extent alleviate the domain shift. However, they still show a distinct distributional gap between different features, which can be owed to the multimodal misalignment in target domain. Obviously, our method progressively generates truly aligned video-text pairs in target domain.

**Qualitative Results.** In Figure 6, we further visualize the returned results across different settings and baselines. Specifically, we show how the rank of relevant videos changes with different methods when given a query text. It is clear that our method leads to higher ranks of relevant videos compared to Source Only and ACP. One limitation is that our method performs worse given the query '*There is lady is riding through the sea waves*', i.e., our method returns the relevant video with rank 10 while ACP with rank 8. One possible reason might be the high similarity of RGB color between foreground lady and background sea waves.

## 5   Conclusion

In this paper, we focus on the notoriously challenging task, i.e., UDA Video-text Retrieval (UDAVR), which can be classified as a cross-modal cross-domain retrieval task. To tackle the *one-to-many* issue, we propose the Uncertainty-aware Alignment Network (UAN), which progressively generates truly aligned target pairs and ensures the discriminability of target features. Extensive experiments and ablation studies across several benchmarks justify the superiority of our method.

**Limitations and Societal Impacts.** One limitation is the difficulty in handling the one-to-many issue at the early training stage. As the massive data (both videos and texts) in target domain have no semantic relations with each other, especially in the first few epochs, it is probably inaccurate when selecting the truly positive pairs. Besides, a simple and effective UDA Video-Text Retrieval

(UDAVR) method is crucial for reduce the huge performance gap between source and target data. We hope that our method can pave the way for effective and robust video-text retrieval system.

## 6  Acknowledgement

This work was supported by the National Key R&D Program of China under Grant 2022YFB3103500, the National Natural Science Foundation of China under Grant 62202459, and the China Postdoctoral Science Foundation under Grants 2022M713348 and 2022TQ0363.

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
