# OpenReview forum: "Uncertainty-Aware Alignment  Network  for Cross-Domain Video-Text Retrieval"
_NeurIPS.cc/2023/Conference — NeurIPS 2023 poster_

### Official Review · Reviewer_FYKf · 2023-07-03

**Soundness:** 3 good
**Presentation:** 4 excellent
**Contribution:** 3 good
**Rating:** 6
**Confidence:** 5

**Summary:**

This paper focuses on text-video retrieval for the challenging unsupervised domain adaptation setting. For this problem, the model is trained on a source only set of supervised video/text pairs and is adapted to a target domain which consists of videos and text labels with no pairing ground truth. The proposed method, named Uncertainty-aware Alignment Network (UAN), uses the assumption that many videos can correspond to a textual item and vice-versa, which allows for the multi-granularity modelling. They evaluate their method on three different pairs of datasets between MSR-VTT, MSVD, and TGIF. Their method is found to outperform all other unsupervised domain adaptation methods for video retrieval because they adapt classifier domain adaptation approaches.

**Strengths:**

The experiments that have been run are convincing and thorough and even include evaluation of the method for image-text domain adaptation which is nice to see and proves that the proposed approach was created from the ground up for embedding spaces instead of borrowed from unsupervised domain adaptation approaches for classification. As a final note, the experiments were all run 5 times and averaged for robustness.

The proposed model works well for unsupervised domain adaptation video retrieval and makes sense to break from the one-to-one assumption during the training to better find potential positives.

**Weaknesses:**

What are the values of a and b given in equation 7/8 and how are these chosen?

Equations 10 and 11 could be better introduced/explained, additionally, it's easy to miss that T and K refer to the batch and for an individual example respectively.

The discussion on failure cases of the model within the paper is rather short. There is one given failure case within figure 6 which is very briefly explained (so much so that I'm not sure how the model failed on this case). I think these could be better highlighted as another future avenue for related work and better understanding of the method as a reader.

The method calls for many-to-one and one-to-many relationships between the two modalities during training, but for evaluation it is assumed that there is only a one-to-one relationship between modalities as with previous work. There has been some work that mentions the possibility of many-to-many relationships during training for images [a] and videos [b] and it might be worth discussing this point within the limitations which also mentions the one-to-many relationships at the early training stage.

[a] Chun, Sanghyuk, et al. "Probabilistic embeddings for cross-modal retrieval." Proceedings of the IEEE/CVF Conference on Computer Vision and Pattern Recognition. 2021.
[b] Wray, Michael, Hazel Doughty, and Dima Damen. "On semantic similarity in video retrieval." Proceedings of the IEEE/CVF Conference on Computer Vision and Pattern Recognition. 2021.

Other Comments
Line 6: missing space between comma and 'as'
[24] and [25] are duplicated citations
Line 142: inner production -> inner product

**Questions:**

1. Can the failure cases of the method be expanded and more information given regarding the failure case in Figure 6
2. What are the values of a and b given in equation 7/8 when training the model and how are these chosen?
3. With the domain adaptation requiring the one-to-one relationship between video and text during evaluation to be relaxed into a many-to-many relationship during training, do you think this has a method on overall method performance?

**Limitations:**

Limitations are discussed within the paper in the final paragraph, however, not much is said for the method beyond selecting pairs in the first few epochs likely causes issues. I think this could be expanded further to round out the paper.

---

> ### Author Rebuttal · Authors · 2023-08-07
>
> Thank you for reviewing our paper and for the positive feedback, pointing out that our method was created `FROM THE GROUND UP FOR EMBEDDING SPACES'! We will revise the typos in the final version.
>
> **Question1**: values of a and b in Eq. 7 and symbols in Eq. 10-12.
>
> **Response1**: In Eq. 7-8 of D-VLDA, a is a negative scale factor since similarity is inversely proportional to the distance, and b is a shift value.
> In Eq. 10-12, ${S}^{\mathcal{V} \rightarrow \mathcal{T}}$ denotes the calculated similarities of target $v_{i}^t$ with all the target texts and similar definition for ${S}^{\mathcal{T} \rightarrow \mathcal{V}}$.
> For all our experiments, we set a to -0.005 and b to 6. We will add the discussion and ablation of parameters a and b in the Experimental section.
>
> **Question2**: discussions of failure cases.
>
> **Response2**: Fig. 6 shows that ours achieves consistently best results compared with other SOTAs. We also report a failure case where 'There is lady is riding through the sea waves'. This is acceptable and can be owed to (1) the low visual quality and short length of this video; (2) the confusion of the front object 'lady' and the background 'sea waves'. One possible reason for the confusion is that the 'lady' target is too small and may be 'swallowed' by the `waves'. As to the future work, we will further add explanations to make it easier for readers to understand our work.
>
> **Question3**: one-to-many in training vs. one-to-one in testing.
>
> **Response3**: We consider the training with a more relaxed one-to-many relationship, which is rather straightforward since one video may have high similarity with more than one text. As for 'one video to many texts', this is widely existed in video-text retrieval datasets. While for 'one text to many videos', this is also intuitive that a textual sentence ($t_1$) may sometimes describe the contents of two similar videos ($v_1$ and $v_2$), even if the ground-truth pair is ($v_1$, $t_1$). This has been empirically identified in previous works like [a] and [b].
> The insight of our method is to fully exploit this one-to-many relationship during training for a better video/text embedding, and this contributes to the overall retrieval performance in target domain.
>
> [a] Bogolin, Simion-Vlad, et al. "Cross modal retrieval with querybank normalisation." Proceedings of the IEEE/CVF Conference on Computer Vision and Pattern Recognition. 2022.
>
> [b] Wray, Michael, Hazel Doughty, and Dima Damen. "On semantic similarity in video retrieval." Proceedings of the IEEE/CVF Conference on Computer Vision and Pattern Recognition. 2021.

---

> > ### Comment · Reviewer_FYKf · 2023-08-17
> >
> > Thank you for clarifying questions from my initial review. I am still in favour of this paper after reading the other reviews and responses and am still leaning towards acceptance of this paper.

---

> > > ### Author Response · Authors · 2023-08-18
> > >
> > > Thanks for your timely reply and positive feedback.
> > > We will revise the final version according to these constructive discussions.
> > > Thanks again and feel free to inform us for any further questions and discussions.

---

### Official Review · Reviewer_qpQ6 · 2023-07-04

**Soundness:** 3 good
**Presentation:** 3 good
**Contribution:** 3 good
**Rating:** 6
**Confidence:** 5

**Summary:**

The paper tackles the task of unsupervised domain adaptation for text-video retrieval. The authors propose an Uncertainty-aware Alignment Network (UAN), which exploits the semantic information of both modalities in target domain. Specifically, in order to tackle the one-to-many relationships in the target domain, the proposed Uncertainty-aware Alignment Mechanism (UAM) tries to utilize the multi-granularity relationships between each target video and text to ensures the discriminability of target features. Finally, the authors test their proposed method on several benchmarks.

**Strengths:**

The paper tackles an important problem and obtains good results. The results can be of high interest for the research community in order since the method seems to be relatively easy to combine with other existing methods, however I think some more clarifications are needed on this front.

**Weaknesses:**

I think some parts of the paper need more explanations (see questions below). Also, for completeness, I think it would be useful to also briefly describe what happens at inference.

Missing papers in text-video retrieval related work

[1] Gorti, Satya Krishna, et al. "X-pool: Cross-modal language-video attention for text-video retrieval." Proceedings of the IEEE/CVF conference on computer vision and pattern recognition. 2022.

[2] Bogolin, Simion-Vlad, et al. "Cross modal retrieval with querybank normalisation." Proceedings of the IEEE/CVF Conference on Computer Vision and Pattern Recognition. 2022.

[minor] typo line 155 "to to"

**Questions:**

1. How are the results for CE reported for text-image retrieval since as far as I know the method only reports results on text-video retrieval? Do you re-train the method on images? What input features do you use for images?

2. Same question for methods in Tab 5? Do you evaluate using the published weights? Re-train? Can an already trained model be adapted to a different domain without re-training? More details are needed

3. Can you give more details on what is needed to combine the proposed method with other existing methods? What's the computational overhead? Are there any additional costs during inference? etc.

**Limitations:**

Limitations are discussed thorough the paper and societal impact is also discussed briefly.

---

> ### Author Rebuttal · Authors · 2023-08-07
>
> Thank you for reviewing our paper and for the positive feedback. We will add the missing papers in the final version.
>
> **Question1**: details of training CE method for image-text retrieval task.
>
> **Response1**: For a fair comparison, we use the same settings for UDA image-text retrieval in the follow paper ACP [a]. Specifically, we use the object, scene and face features in the image to re-train the model in the CE [b] method.
>
> [a] Y. Liu, Q. Chen, and S. Albanie. Adaptive cross-modal prototypes for cross-domain visual-language retrieval. In Proceedings of the IEEE/CVF Conference on Computer Vision and Pattern Recognition, pages 14954–14964, 2021.
>
> [b] Y. Liu, S. Albanie, A. Nagrani, and A. Zisserman. Use what you have: Video retrieval using representations from collaborative experts. arXiv preprint arXiv:1907.13487, 2019.
>
> **Question2**: details of the combination of our method with existing methods.
>
> **Response2**: In Tab. 5, we combine the proposed method with other methods by re-training the original methods according to their source codes. For example, the method of `HGR + UAN' is obtained by re-training HGR with the addition of our D-VLDA and UAM modules. This leads to a fair comparison.
>
> **Question3**: details of computation overhead and inference procedure.
>
> **Response3**: During the training phase, the calculation of pair similarities takes about 0.006s for each batch, which is reasonable and acceptable due to the dataset scale and batch size. In the inference stage, we directly use the model trained on the source domain and test it on the target domain directly, which does not involve any computational overhead.

---

> > ### Comment · Reviewer_qpQ6 · 2023-08-17
> >
> > Thanks for the clarification. If the authors will add the clarifications in the final version of the paper, I tend towards acceptance.

---

> > > ### Author Response · Authors · 2023-08-18
> > >
> > > Thanks for your timely reply and positive feedback.
> > > We will update these valuable discussions and clarifications in the final version.
> > > Thanks again and feel free to inform us for any further questions and discussions.

---

### Official Review · Reviewer_dcEM · 2023-07-06

**Soundness:** 3 good
**Presentation:** 2 fair
**Contribution:** 3 good
**Rating:** 5
**Confidence:** 4

**Summary:**

This paper proposed a method named Uncertainty-aware Alignment Network (UAN) to address the one-to-many issue which means in the real scene, one text usually corresponds to multiple videos and vice versa. Specifically, the proposed method achieves a new state-of-the-art in Cross-Domain Video-Text Retrieval through the incorporation of design elements such multi-modal mutual information module and Uncertainty-aware Alignment Mechanism (UAM).

**Strengths:**

1.	The motivation for addressing “one-to-many in target domain” is clearly presented and validated.
2.	The ablation study and generalization experiment in this paper is comprehensive and clearly analyzed, providing strong evidence of the validity of the proposed model.


**Weaknesses:**

1.	It is recommended to add baseline performance in Fig 4 to reflect the robustness of the proposed module.
2.	The authors used each batch in the Uncertainty-Aware Alignment Mechanism to obtain self-discovered matching pairs instead of all data, so will the different number of batch have a significant impact on the effectiveness?
3.	The phrase "the confusion of the lady and sea waves" mentioned in the Qualitative Results section is ambiguous. Further analysis is recommended in this section to clarify this point.

**Questions:**

N/A

---

> ### Author Rebuttal · Authors · 2023-08-07
>
> Thank you for reviewing our paper and for the positive feedback.
>
> **Question1**: add baseline in Fig. 4.
>
> **Response1**: Thanks for your suggestion, we will add baseline performance in Fig. 4 to reflect the robustness of the proposed module in the camera version of the paper.
>
> **Question2**: ablation on the number of batch size.
>
> **Response2**: Thanks for the advice. This is a very interesting ablation study, and we report the results as follows. As can be seen, when the batch size is from 16 to 64, the performance tends to increase. However, the performance of the batch size from 64 to 256 tends to be stable. This interesting ablation is unexplored before, and we argue that this may be partially due to that small batch size will exclude some ground-truth pairs, while large batch size will lead to the computational overhead instead of more included positive pairs. We will add discussion and analysis in the camera version.
>
> |**Batch** | | Tf->Mt |  |  | Mt->Tf |  |
> |  :----:   |    :----:   |  :---: | :---: | :---: | :---: | :---: |
> |   | R@1 | R@10| MR | R@1 | R@10| MR |
> | 16     | 4.22    | 22.36   | 79| 7.41|32.03|33
> | 32   | 5.62   | 25.56   | 45| 8.36|29.61|41
> | **64**   | **6.12**    | **27.23**   |**40**| **9.16**|**31.06**|**37**
> | 128   | 6.05 | 27.13  | 40| 9.04|30.87|37
> | 256    | 6.03 | 27.08   | 40| 9.01|30.72|37
>
>
>
>
> **Question3**: the phrase "the confusion of the lady and sea waves".
>
> **Response3**: Fig. 6 shows that ours achieves consistently best results compared with other SOTAs. We also report a failure case where 'There is lady is riding through the sea waves'. This is acceptable and can be owed to (1) the low visual quality and short length of this video; (2) the confusion of the front object 'lady' and the background 'sea waves'. One possible reason for the confusion is that the 'lady' target is too small and may be 'swallowed' by the `waves'. As to the future work, we will further add explanations to make it easier for readers to understand our work.

---

### Official Review · Reviewer_muWC · 2023-07-06

**Soundness:** 3 good
**Presentation:** 2 fair
**Contribution:** 2 fair
**Rating:** 5
**Confidence:** 4

**Summary:**

This paper proposed a unsupervised domain adaptation method tailored for video-text retrieval by exploring the one-to-many correspondences between video and text on the target domains. The proposed method is shown effective and superior to existing approaches which considered only the one-to-one video-text matching relationships on the unlabelled target domains.

**Strengths:**

+ The idea to learn from the one-to-many matching relationships among video and text in real-world scenarios is intuitive, the proposed model is straighforward
+ The proposed method not only yielded impressive performance on video-text retrieval, the authors further demonstrate its beneficial to image-text retrieval. Moreover, the authors also proved that the proposed idea is generally beneficial to the task by combining it with a wide range of existing methods.

**Weaknesses:**

+ The motivations are not sufficiently discussed. The two main motivations claimed in the paper is (1) existing UDAVR approaches are mostly derived from classification based DA methods which is not optimal for retrieval tasks and (2) existing methods assume a one-to-one matching relationships between video and text. For (1), it may be more straightforward to spell out the major drawbacks of adapting classification DA methods to UDAVR rather than just stating it is suboptimal. The motivation (2) should be justified in terms of whether the one-to-many matching relationships are common on the adopted training datasets. It will be more intuitive if this can be quantified.
+ I'm not sure I fully understand what is the meaning of "to balance the minimization of domain shift" at L12
+ As stated at L36, the definition of UDAVR is that there is no identical textual labels on the two domains. However, two sentences even from the same domains are not likely to be exactly identical but they can describe the same events using a similar vocabulary. Therefore, does it make more sense to measure how different are two video-text retrieval datasets by the overlaps of their vocabularies?
+ In Fig.1, I failed to find the differences between existing methods which are claimed to be classification-based and the proposed method designs for retrieval tasks (L55)
+ In Eq.(7)-(9), as a "negative scale factor", $a$ seems to be a constant, then how about $b$? is it also a constant? If yes, then why adding a constant in a loss function to be optimised?
+ Are the symbols $S^{T\rightarrow V}$ and $S^{V\rightarrow T}$ used in Eq.(10) and Eq.(11) defined?
+ minor problems (typos and etc):
  - duplicated "to" at L12, L155
  - "similarity Top-K" in Eq.(12)

**Questions:**

+ It will be interesting to see how many discovered one-to-many matching relationships are true and consistent with the manual labels on a validation set.
+ In Fig.4, the model obtained its best performance when K=2, does this mean that each video is optimally to be matched with two text sentences? This seems to be applied to all the pairwise combination of datasets but it is not intuitive why a 1-to-2 mapping is always the best. adding more discussion and analysis might provide more insights.

**Limitations:**

Yes, the author discussed one of the limitations of their proposed method is the false positive inter-sample relationships discovered for training, especially at the early stage.

---

> ### Author Rebuttal · Authors · 2023-08-07
>
> Thank you for reviewing our paper and for the very constructive feedback. We will revise the typos in the final version.
>
> **Question1**: drawbacks of directly adapting classification-based DA methods in UDAVR task.
>
> **Response1**: Previous approaches are mostly derived from classification based domain adaptation methods, which are neither multi-modal nor suitable for retrieval task.
> Intuitively, (1) classification-based DA methods are designed for single-modality (like image classification) task, which can't be directly applied in multi-modal setting (UDAVR task); (2) their main assumption is the common label set for source and target domains, which still can't be satisfied in UDAVR task. Empirically, in Tab. 4, we conduct extensive ablation studies with several state-of the-art domain adaptation methods. The proposed D-VLDA achieves significant improvements compared with classification-based and conditional alignment DA methods, which indicates that the D-VLDA module plays an essential role in the UDA video-text retrieval task.
>
> **Question2**: the commonness of one-to-many.
>
> **Response2**: Thanks for the constructive advice. We conduct ablations to verify this as follows. We randomly select 1,000 video-text pairs and calculate the semantic similarities between each two texts (with details in [a]). We then define one video and one text as a `one-to-many' pair if the corresponding similarity is above the threshold. Results can be found below. The one-to-many relationship contributes to the selected target pairs for training and the overall performance.
>
>
> |**Dataset**|S=1.0 | S>0.9 |S>0.8  | S>0.7 |
> |  :----:   |    :----:   |  :---: | :---: | :---: |
> |MSR-VTT|1,000|1,232|1,398|1,478|
> |MSVD|1,000|1,268|1,426|1,520|
> |TGIF|1,000|1,198|1,306|1,406|
>
>
>
>
> [a] Wray, Michael, Hazel Doughty, and Dima Damen. "On semantic similarity in video retrieval." Proceedings of the IEEE/CVF Conference on Computer Vision and Pattern Recognition. 2021.
>
> **Question3**: the meaning of "to balance the minimization of domain shift".
>
> **Response3**: Different datasets usually have inconsistent data distributions and representations, thus leading to the domain shift problem.
> To alleviate this, we propose D-VLDA module to balance the minimization of domain shift on the source and target domains.
> By `balance' we refer to the dynamically minimized domain gap designed for multi-modal setting.
>
> **Question4**: measure the difference of two video-text retrieval datasets by the overlaps of their vocabularies.
>
> **Response4**: Thanks for the advice. We admit that in some video-text retrieval datasets, one video may have high similarity with more than one text. For instance, in MSR-VTT, each video corresponds to 20 texts. Besides, the video may also have relatively high similarity with other texts (which belong to other videos). However, we argue that in UDAVR task, the domain gap mainly denotes that there exist large differences like video sources (cartoons, movies, sports etc.), text sources (short sentences, movie dialogues, cooking instructions etc.), video/text lengths and so on. THIS defines the uniqueness of UDAVR task.
>
> **Question5**: illustration of Fig. 1.
>
> **Response5**: Thanks for your suggestion, we will further optimize our Fig. 1 in the camera version.
> In Fig. 1, existing works are mostly derived from classification based domain adaptation methods to align different domains.
> However, we propose Distribution-based Vision-Language Domain Adaptation, called D-VLDA, to relieve the divergence of domain statistics, thus the distribution shifts can be significantly diminished, improving the generalization of learned model on out-of-distribution target domain.
>
> **Question6**: details of Eq. 7-8.
>
> **Response6**: In Eq. 7-8 of D-VLDA, a is a negative scale factor since similarity is inversely proportional to the distance, and b is a shift value. For all our experiments, we set a to -0.005 and b to 6. We will add a discussion of parameters a and b in the Experimental Details section.
>
> **Question7**: why a 1-to-2 mapping is always the best.
>
> **Response7**: By '1-to-2' relationship we denote that during training time, one video can be considered to be `paired' with more than one text (the ground-truth one) and vice versa. We claim that training with one-to-many benefits the target retrieval performance even when evaluating with one-to-one setting. We will further clarity this in the final version.

---

> > ### Author Response · Authors · 2023-08-18
> >
> > Thanks again for your constructive reviews and we are looking forward to further discussions.
> > As to the "one-to-many" issue, which is also mentioned in Reviewer NzVt, we add the following clarifications as further response.
> >
> > **Q**: how many ground-truth pairs are among the newly found pairs by top-2 UAM
> >
> > **A**: We show the results of setting TGIF dataset to MSRVTT dataset in Table 1.
> > For instance, 3/18(16.66\%) denotes 18 pairs are selected out, of which 3 pairs are ground-truth pairs, forming a ratio of 16.66\%. Clearly, UAM selects out 6 more pairs, of which 2 are ground-truth ones.
> > Besides, as training proceeds, the ratio of ground-truth pairs with UAM also increases and are consistently larger than that of DAC.
> > This is reasonable and consistent with our intuition that more selected pairs (including 2/6 ground-truth pairs and 4/6 pseudo aligned pairs with high similarities) will benefit the retrieval performance in target domain.
> > |**Epoch**|20 | 40 |60 | 80 | 100 |
> > |  :----:   |    :----:   |  :---: | :---: | :---: | :---: |
> > |UAN(W/ DAC)|3/18(16.6\%)|8/23(34.78\%)|10/24(41.66\%)|11/28(39.28\%)|13/31(41.93\%)|
> > |UAN(W/ UAM)|5/24(20.83\%)|12/32(37.5\%)|15/34(44.11\%)|17/37(45.94\%)|21/34(48.33\%)|
> > |Newly selected|2/6(33.33\%)|4/9(44.44\%)|5/10(50\%)|6/11(54.54\%)|8/13(61.53\%)|
> >
> >
> > Thanks again and feel free to inform us for any further questions and discussions.

---

> > ### Comment · Reviewer_muWC · 2023-08-18
> >
> > I thank the authors for their efforts in rebuttal. I have checked their responses as well as the comments from other reviewers carefully. Some of my concerns have been addressed. However, I still believe that the key assumption held by this method should be further verified to make the paper stronger. The authors provided additional statistics to show that there are around 20% (with a threshold of 0.9) more video-text pairs are matched in semantics but overlooked in the manual annotations. The key question here to me is how will this affect existing UDAVR solutions, which is under-explored. For example, if we dropped all the video/text involving one-to-many matching relationships on training data, how will this affect the proposed method or existing models that are claimed to suffer from the assumption of one-to-one matching? Such an investigation can be helpful in explaining why the proposed ideas are effective.  I also agree with Reviewer#NzVt that additional examples of the mined video-text pairs which were initially missing in the manual labels will help with understanding, as well as the discussions about whether such one-to-many relationships should be considered in the source domains or testing.
> >
> > In summary, although the proposed method is intuitive and effective to me as commented pre-rebuttal, I think the most critical part about the assumption still needs to be improved. So I would keep my initial rating.

---

> > > ### Comment · Reviewer_FYKf · 2023-08-18
> > >
> > > Hi muWC,
> > > Thanks for your comment, I'm currently positive towards this paper and after reading your response I didn't quite follow a couple of points that you brought up.
> > >
> > > > For example, if we dropped all the video/text involving one-to-many matching relationships on training data, how will this affect the proposed method or existing models that are claimed to suffer from the assumption of one-to-one matching?
> > >
> > > The authors argue that in this paper existing models are assuming a one-to-one for their benefit, which has also been found out in multiple papers [a], [b]. From [b] we know that common datasets (of which MSR-VTT is used in this paper) have this issue of one-to-many. Both Table 3 and Table 5 in the original paper showcase the effects of removing/adding the one-to-many matching and testing their system on top of other VR approaches.
> > >
> > > > The key question here to me is how will this affect existing UDAVR solutions, which is under-explored.
> > > Other approaches use DA approaches which are based on classification approaches. I'm not sure how the one-to-many would be applied to these other approaches beyond applying UAN on top of them.
> > >
> > > Would you be able to expand on your reasoning here?
> > >
> > > Thanks
> > >
> > > [a] Chun, Sanghyuk, et al. "Probabilistic embeddings for cross-modal retrieval." Proceedings of the IEEE/CVF Conference on Computer Vision and Pattern Recognition. 2021.
> > >
> > > [b] Wray, Michael, Hazel Doughty, and Dima Damen. "On semantic similarity in video retrieval." Proceedings of the IEEE/CVF Conference on Computer Vision and Pattern Recognition. 2021.

---

> > > > ### Author Response · Authors · 2023-08-18
> > > >
> > > > Dear Reviewers FYKf and muWC:
> > > >
> > > > We highly appreciate the constructive discussions and we will response as follows.
> > > >
> > > > (1)As to the first question of "how the one-to-many affects the performance", we TOTALLY agree with what Reviewer FYKf has clarified.
> > > > Table 3 of this paper shows the results of adding or removing UAM module, which exactly indicates the effectiveness of one-to-many.
> > > >
> > > > (2)As to the second question of "how exactly one-to-many is combined with other video-text retrieval methods", we might quote that (which is also raised by Reviewer qpQ6) "In Tab. 5, we combine the proposed method with other methods by re-training the original methods according to their source codes. For example, the method of `HGR + UAN' is obtained by re-training HGR with the addition of our D-VLDA and UAM modules. This leads to a fair comparison."
> > > >
> > > > We sincerely thank for both the positive attention of Reviewer FYKf and the valuable concerns of Reviewer muWC, and we are looking forward for your replies.

---

> > > > > ### Comment · Reviewer_muWC · 2023-08-18
> > > > >
> > > > > Thanks FYKf and the authors for the further clarifications. I have no doubt about the effectiveness of UAM. The missing part I tried to point out is the investigation of how important the matching uncertainty studied in this paper is to limiting the performance of existing UDAVR methods that hold the one-to-one assumption. In other words, will existing models (eg DAC [19]) perform more poorly/better if there were more/less one-to-many matching pairs in training data? This is about the motivation of UAM (similar to the concern on how common is the one-to-many issue), rather than its effectiveness which has been well proven by the provided experimental results. Although the effectiveness of an idea is usually relevant to whether it is well-motivated, explicit justification of the motivation will make it more straightforward to understand the performance advantages obtained. Given that the motivation of UAM seems to be intuitive to most of the other reviewers, I'll re-consider how necessary is it to further justify this.

---

> > > > > > ### Author Response · Authors · 2023-08-19
> > > > > >
> > > > > > Dear Reviewer muWC:
> > > > > >
> > > > > > We sincerely thank for your continuous suggestions. As we quote from Reviewer NzVt that "In my understanding, the Uncertainty-Aware Alignment Mechanism module can be seen as a pseudo-label generator that is used to find matching video text pairs in the target domain. Even if the "one-to-many" assumption does not hold, it seems that the “top-2 UAM” algorithm helps to find more true-positive pairs (albeit with a potential for increased noise), thus allowing the model to learn more on target domain."
> > > > > > **This is EXACTLY our motivation and we hope this may help to the understanding.**
> > > > > >
> > > > > > Moreover，in order to verify our motivation, we further conduct the following experiments:
> > > > > >
> > > > > > Q1: how many ground-truth pairs are among the newly found pairs by top-2 UAM
> > > > > >
> > > > > > A1: We show the results of setting TGIF dataset to MSRVTT dataset in Table 1.
> > > > > > For instance, 3/18(16.66\%) denotes 18 pairs are selected out, of which 3 pairs are ground-truth pairs, forming a ratio of 16.66\%. Clearly, UAM selects out 6 more pairs, of which 2 are ground-truth ones.
> > > > > > Besides, as training proceeds, the ratio of ground-truth pairs with UAM also increases and are consistently larger than that of DAC.
> > > > > > This is reasonable and consistent with our intuition that more selected pairs (including 2/6 ground-truth pairs and 4/6 pseudo aligned pairs with high similarities) will benefit the retrieval performance in target domain.
> > > > > > **These results INTUITIVELY JUSTIFY our motivation and will be added to the final version.**
> > > > > >
> > > > > >
> > > > > > |**Epoch**|20 | 40 |60 | 80 | 100 |
> > > > > > |  :----:   |    :----:   |  :---: | :---: | :---: | :---: |
> > > > > > |UAN(W/ DAC)|3/18(16.6\%)|8/23(34.78\%)|10/24(41.66\%)|11/28(39.28\%)|13/31(41.93\%)|
> > > > > > |UAN(W/ UAM)|5/24(20.83\%)|12/32(37.5\%)|15/34(44.11\%)|17/37(45.94\%)|21/34(48.33\%)|
> > > > > > |Newly selected|2/6(33.33\%)|4/9(44.44\%)|5/10(50\%)|6/11(54.54\%)|8/13(61.53\%)|
> > > > > >
> > > > > >
> > > > > > Thanks again and we will definitely re-arrange all these valuable discussions and add them to the final version.

---

> > > > > > > ### Comment · Reviewer_muWC · 2023-08-21
> > > > > > >
> > > > > > > Many thanks to the authors for their tremendous efforts in explaining their justification of motivation. Given that matching uncertainty has been shown to be common both by the authors and in existing studies, further exploration of the scenarios where one-to-many matching relationships don't exist may not be practical. I'm happy to change my rating to a 5 and encourage the authors to incorporate those clarifications and discussions in their final version.

---

> > > > ### Author Response · Authors · 2023-08-19
> > > >
> > > > We sincerely thank again for your help of clear explanations and we have added more experimental results to help Reviewer muWC's understanding.

---

### Official Review · Reviewer_NzVt · 2023-07-07

**Soundness:** 3 good
**Presentation:** 4 excellent
**Contribution:** 4 excellent
**Rating:** 5
**Confidence:** 5

**Summary:**

This paper addresses the unsupervised domain adaptation video-text retrieval problem by two proposed components, the D-VLDA (Distribution-based Vision-Language Domain Adaptation) and the UAM (Uncertainty-Aware Alignment). The proposed D-VLDA aims at alleviating the domain discrepancy via moment-based method, while the UAM method aims for better pseudo labeling.

**Strengths:**

(+) This article is well written, and the description of the method pipeline is clear and easy to understand.

(+) Experiments are sufficient and ablation studies show the effectiveness of the proposed components.

(+) Stat-of-the-art results are demonstrated on several datasets.

**Weaknesses:**

(-) Unfair comparison in table 3, see in Question(2).

(-) The "one-to-many" assumption that underpins the article lacks empirical evidence supporting its widespread occurrence.

**Questions:**

(1) The term "(moment-based) D-VLDA module" appears to be referred to as "multi-modal mutual information module" in the abstract. Could the author please clarify the relationship between these two concepts?

(2) In the design of Uncertainty-Aware Alignment(UAM), the authors expand the top-1 dual alignment method from [A] to a top-k dual alignment. It appears that Table 3 aims to validate this expansion, where "UAN(w/DAC)" denotes the application of the top-1 dual alignment consistency mechanism to pick those most similar videos and texts in the target domain, as described in [A]. However,  I note that the results for "UAN(w/DAC)" match those reported in [A], while [A] utilizing a different discrepancy loss from D-VLDA. This may lead to an unfair comparison. I would suggest the authors control the variables to ensure a more equitable comparison.

(3) The authors propose the uncertainty-aware alignment network based on the "one-to-many" assumption: one text can describe multiple videos and vice versa. This forms the fundamental premise of the paper. However, the frequency of the "one-to-many" phenomenon remains unclarified. I'm concerned that the efficacy of the method may not be primarily due to the "one-to-many" assumption, but potentially other factors like the top-K methods introduce more true positive pairs in the early training stage, providing the model with more supervision about the target domain. I think the author should verify the frequent existence of the "one-to-many". For example, collecting the top-K dual alignment results by the final model, removing the pairs that belong to the ground truth pairs, and then see the proportion of the remaining pairs to all the ground truth pairs, and also estimate how many of the left pairs match correctly (belongs to the "one-to-many").

[A] X. Hao, W. Zhang, D. Wu, F. Zhu, and B. Li. Dual alignment unsupervised domain adaptation for video-text retrieval. In Proceedings of the IEEE/CVF conference on computer vision and pattern recognition, 2023.

---

> ### Author Rebuttal · Authors · 2023-08-07
>
> Thank you for reviewing our paper and for the positive feedback.
>
> **Question1**: unfair comparison in Tab. 3.
>
> **Response1**: Thank you for pointing out this, and the corrected experimental data is as follows:
>
> |**Method** | | Tf->Mt |  |  | Mt->Tf |  |
> |  :----:   |    :----:   |  :---: | :---: | :---: | :---: | :---: |
> |   | R@1 | R@10| MR | R@1 | R@10| MR |
> | UAN(w/ DAC)  | 5.43    | 25.63   | 48| 8.44|29.12|42
> | UAN(w/ UAM)  | 5.76  | 26.12   | 43| 8.73|29.84|40
> | **UAN(full)**   | **6.12**    | **27.23**   |**40**| **9.16**|**31.06**|**37**
>
>
>
> **Question2**: "(moment-based) D-VLDA module" vs. "multi-modal mutual information module".
>
> **Response2**: Thanks for the advice. We claim that by 'multi-modal' we refer to the contribution of our proposed domain alignment module, which is the first in UDAVR to tackle the domain gap with the consideration of different modalities. While `moment-based D-VLDA' denotes the specific technique we adopted (minimizing the domain gap in a distributional manner). We will align these terms for more clarity in the final version.
>
>
> **Question3**: the frequent existence of the "one-to-many".
>
> **Response3**: Thanks for the constructive advice. We conduct ablations to verify this as follows. We randomly select 1,000 video-text pairs and calculate the semantic similarities between each two texts (with details in [a]). We then define one video and one text as a `one-to-many' pair if the corresponding similarity is above the threshold. Results can be found below. The one-to-many relationship contributes to the selected target pairs for training and the overall performance.
>
> |**Dataset**|S=1.0 | S>0.9 |S>0.8  | S>0.7 |
> |  :----:   |    :----:   |  :---: | :---: | :---: |
> |MSR-VTT|1,000|1,232|1,398|1,478|
> |MSVD|1,000|1,268|1,426|1,520|
> |TGIF|1,000|1,198|1,306|1,406|
>
> [a] Wray, Michael, Hazel Doughty, and Dima Damen. "On semantic similarity in video retrieval." Proceedings of the IEEE/CVF Conference on Computer Vision and Pattern Recognition. 2021.

---

> ### Comment · Reviewer_NzVt · 2023-08-17
>
> I thank the authors for their careful replay. My first two questions have been addressed. I still concern about the basic assumption the frequent existence of "one-to-many" as mentioned in my question (3). I can see the same concern from Reviewer muWC.
>
> In my understanding, the Uncertainty-Aware Alignment Mechanism module can be seen as a pseudo-label generator that is used to find matching video text pairs in the target domain. Even if the "one-to-many" assumption does not hold, it seems that the “top-2 UAM” algorithm helps to find more true-positive pairs (albeit with a potential for increased noise), thus allowing the model to learn more on target domain, while the basic “top-1 UAM” algorithm results in many true-positive pairs being filtered out.
>
> (1) In the authors' reply,  the authors calculate the similarities among 1000 text and pick more than 200 new pairs whose similarities are larger than 0.9. Can the authors show some pairs they found. By the way, it is different from how UAM finds multiple text pairs belonging to the same video, so I am still confused about how many new pairs will be found by the UAM. Top-2 UAM algorithm will obviously finds more pairs than top-1 UAM, so in the newly found pairs, how many of them belongs to the ground-truth pairs.
>
> (2) If "one-to-many" exists frequently in the datasets, does that mean the UAM should also be applied in the source domain.
>
> The Uncertainty-Aware Alignment Mechanism module does show effectiveness. My concern relates to the author's explanation about why it is effective.

---

> > ### Author Response · Authors · 2023-08-18
> >
> > Thanks for your timely reply and constructive feedback.
> > We clarify your concerns as follows.
> >
> > **Q1**: how many ground-truth pairs are among the newly found pairs by top-2 UAM
> >
> > **A1**: We show the results of setting TGIF dataset to MSRVTT dataset in Table 1.
> > For instance, 3/18(16.66\%) denotes 18 pairs are selected out, of which 3 pairs are ground-truth pairs, forming a ratio of 16.66\%. Clearly, UAM selects out 6 more pairs, of which 2 are ground-truth ones.
> > Besides, as training proceeds, the ratio of ground-truth pairs with UAM also increases and are consistently larger than that of DAC.
> > This is reasonable and consistent with our intuition that more selected pairs (including 2/6 ground-truth pairs and 4/6 pseudo aligned pairs with high similarities) will benefit the retrieval performance in target domain.
> >
> >
> > |**Epoch**|20 | 40 |60 | 80 | 100 |
> > |  :----:   |    :----:   |  :---: | :---: | :---: | :---: |
> > |UAN(W/ DAC)|3/18(16.6\%)|8/23(34.78\%)|10/24(41.66\%)|11/28(39.28\%)|13/31(41.93\%)|
> > |UAN(W/ UAM)|5/24(20.83\%)|12/32(37.5\%)|15/34(44.11\%)|17/37(45.94\%)|21/34(48.33\%)|
> > |Newly selected|2/6(33.33\%)|4/9(44.44\%)|5/10(50\%)|6/11(54.54\%)|8/13(61.53\%)|
> >
> > **Q2**: why not apply "one-to-many" in the source domain
> >
> > **A2**: This is a very constructive suggestion.
> > Indeed, as stated in this paper, "one-to-many" is widely existed in video-text retrieval datasets, which also can be (or should be) tackled in the source domain.
> > However, we intuitively argue (which has also been empirically verified) that solving it in source domain would not contribute to the performance gain in target domain, even if it may benefit the source domain performance.
> > FOR INSTANCE, simply applying top-2 UAM in source domain (TGIF dataset) has a R@1 gain from 8.1 to 9.5, while merely leading to a slight increase on target domain (MSRVTT dataset) from 6.12 to 6.14.
> > This is also somewhat acceptable and reasonable due to large domain gap and modality complexity.
> > We will update the full experimental results in the final version.
> >
> > Thanks again and feel free to inform us for any further questions and discussions.

---

### Author Rebuttal · Authors · 2023-08-07

We would like to thank the reviewers for their efforts, detailed reviews and interest for our submission. We will integrate all their remarks in the revised version of the paper.

---

### Decision · Program_Chairs · 2023-09-21

**Decision:**

Accept (poster)

**Comment:**

This paper proposes  D-VLDA (Distribution-based Vision-Language Domain Adaptation) and the UAM (Uncertainty-Aware Alignment) to address the unsupervised domain adaptation video-text retrieval. Concerns were originally raised regarding unfair comparisons, unclear presentations and baseline results.

 The authors provided a rebuttal and after the discussion period, all reviewers vote for acceptance. The AC recommends acceptance, and encourages the authors to revise paper accordingly.